# Liver Function in Patients with Long-Term Coronavirus Disease 2019 of up to 20 Months: A Cross-Sectional Study

**DOI:** 10.3390/ijerph20075281

**Published:** 2023-03-28

**Authors:** Igor Costa de Lima, Daniel Carvalho de Menezes, Juliana Hiromi Emin Uesugi, Cléa Nazaré Carneiro Bichara, Pedro Fernando da Costa Vasconcelos, Juarez Antônio Simões Quaresma, Luiz Fábio Magno Falcão

**Affiliations:** 1Center for Biological Health Sciences, State University of Pará, Belém 66087670, Brazil; 2Tropical Medicine Center, Federal University of Pará, Belém 66055240, Brazil; 3School of Medicine, São Paulo University, São Paulo 01246903, Brazil

**Keywords:** COVID-19, post-COVID, liver, aspartate aminotransferase, risk factors

## Abstract

The long-term laboratory aspects of the effects of coronavirus disease 2019 (COVID-19) on liver function are still not well understood. Therefore, this study aimed to evaluate the hepatic clinical laboratory profile of patients with up to 20 months of long-term COVID-19. A total of 243 patients of both sexes aged 18 years or older admitted during the acute phase of COVID-19 were included in this study. Liver function analysis was performed. Changes were identified in the mean levels of alanine aminotransferase (ALT), aspartate aminotransferase (AST), lactate dehydrogenase (LDH), gamma-glutamyl transferase (GGT), and ferritin. A ferritin level of >300 U/L was observed in the group that presented more changes in liver function markers (ALT, AST, and GGT). Age ≥ 60 years, male sex, AST level > 25 U/L, and GGT level ≥ 50 or 32 U/L were associated with an ALT level > 29 U/L. A correlation was found between ALT and AST, LDH, GGT, and ferritin. Our findings suggest that ALT and AST levels may be elevated in patients with long-term COVID-19, especially in those hospitalised during the acute phase. In addition, an ALT level > 29 U/L was associated with changes in the levels of other markers of liver injury, such as LDH, GGT, and ferritin.

## 1. Introduction

A significant proportion of patients who survived in the acute phase of coronavirus disease 2019 (COVID-19) reported the continuation or onset of symptoms within 4 or more weeks after the initial infection [1,2]. Such symptoms included fatigue, loss of smell and taste, muscle and joint pain, tachycardia, unexplained decrease or increase in pressure, and shortness of breath [3,4,5,6].

COVID-19 may be associated with liver damage due to direct injury to the liver cells caused by the viral infection [7], delayed resolution of inflammation, and viral persistence [8]. A preliminary study suggested that angiotensin-converting enzyme 2 (ACE2) expression is enriched in the cholangiocytes, and that severe acute respiratory syndrome coronavirus 2 can bind directly to ACE-positive cholangiocytes, thereby deregulating the liver function [9,10]. Cholangiopathy has been reported in three hospitalised patients with long-term COVID-19 who showed significantly high levels of liver markers during their hospital stay and developed new cholangiopathy [11]. Patients with long-term COVID-19 demonstrated persistently increased levels of activated cluster of differentiation (CD) 14+ and CD16+ monocytes, plasmacytoid dendritic cells, and type I and III interferon (IFN-β and IFN-λ1, respectively), levels compared with control individuals within 8 months after infection [12].

Some studies have reported changes in the levels of serum markers in long-term COVID-19 patients, including inflammatory (C-reactive protein (CRP)) and liver markers (alanine aminotransferase (ALT) and aspartate aminotransferase (AST)) [13,14,15,16]. Patients who develop lung lesions (characterised by increased ALT and AST levels) have worse clinical outcomes, such as liver fibrosis and heart disease [17]. Thus, this study aimed to analyse the serum markers of liver damage in patients with up to 20 months of long-term COVID-19.

## 2. Materials and Methods

### 2.1. Type of Study and Ethical Aspects

This cross-sectional, quantitative, descriptive, and analytical observational study followed the criteria adopted by Strengthening the Reporting of Observational Studies in Epidemiology. This study was approved by the Research Ethics Committee of the State University of Pará (opinion no. 4.252.664/2020). All participants provided written informed consent.

### 2.2. Sampling and Study Population

The study participants were selected through convenience sampling from those included in the Long COVID-19 Program in the Brazilian Eastern Amazon between March 2020 and December 2021. A total of 448 adult patients of both sexes, aged 18 years or older, who were not using hepatoprotective or anti-inflammatory drugs during the study, diagnosed with COVID-19 using reverse transcriptase polymerase chain reaction, and who developed at least one long-term symptom related to COVID-19, such as cough, dyspnoea, chest pain, muscle weakness, loss of balance, tremor, fatigue, muscle pain, headache, visual disturbances, insomnia, and/or lower limb oedema, not attributable to another differential diagnosis, for at least 4 weeks after the onset of symptoms, were selected. Of these patients, 134 with incomplete data, who did not undergo blood sample collection (*n* = 34), with incomplete data on laboratory analysis results (*n* = 35), and with history of previous liver disease, including chronic hepatitis B/C virus infection (*n* = 2), were excluded. Thus, only 243 patients were eligible for this study. For data comparison, the patients were allocated into the following groups: (i) long duration of COVID-19 (≤6 months, *n* = 80; >6 months, *n* = 163), (ii) hospitalisation (yes, *n* = 87; no, *n* = 156), (iii) inflammatory markers (all tests were performed for all patients, *n* = 243), and (iv) liver injury markers (all tests were performed for all patients, *n* = 243) (Figure 1). The total, direct, and indirect bilirubin and albumin tests were only performed on 94 patients.

### 2.3. Clinical Data

The following data were collected through interviews and clinical evaluation of the study patients: sex, age, main symptoms presented, and comorbidities.

### 2.4. Liver Function Data

Blood samples were collected in the post-acute phase for biochemical and haematological analyses. The time interval between symptom onset and diagnostic confirmation was 3 days, while the interval between the diagnosis of COVID-19 and the clinical and laboratory evaluation of long COVID patients ranged from 30 to 632 days. The blood samples were stored in tubes containing a clot activator and a separator gel, using the serum for biochemical analysis. To measure the prothrombin time (PT), the citrated samples were subjected to centrifugation for 10 min at 3000 RPM, while the ethylenediaminetetraacetic acid-contaminated samples were used in the haematological analyses.

The blood serum was used (50 μL) in all analyses, except in the measurement of the erythrocyte sedimentation rate (ESR), in which whole blood was used. Biochemical analysis of liver markers was performed using the semiautomatic biochemical analyser CDM 600 (Wiener Lab, Rosario, Argentina).

PT was evaluated using the Humaclot Junior instrument (In Vitro Diagnóstica LTDA, Belo Horizonte, Brazil). Blood serum (25 μL liquid PT) was used in the analysis, enabling visualisation of the test results after 1 min of reaction.

The ESR was determined using graduated pipettes, inserted directly into tubes with anticoagulant, and maintained for 1 h; then, the readings were taken on the descending scale of the tubes. 

The following specific laboratory parameters were measured: ALT, AST, and LDH levels (serum markers of liver injury); bilirubin, alkaline phosphatase (ALP), and GGT levels (commonly associated with cholestasis); albumin (an indicator of liver capacity and synthesis); and PT, ferritin, CRP, and ESR (usually altered in inflammatory processes associated with liver damage). As a reference parameter, the values used by the respective reagent manufacturers were adopted, in accordance with the standards of the laboratory where the analyses were carried out (Appendix A).

### 2.5. Statistical Analysis

The collected data were analysed using GraphPad Prism™ software version 8.4.3 (GraphPad Software, San Diego, CA, USA). The D’Agostino–Pearson test was performed to determine the normality of data distribution, and the results were expressed as mean and standard deviation. The Mann–Whitney and chi-squared tests were used to compare variables without a normal distribution, while analysis of variance (ANOVA) was used for comparisons of variables with a normal distribution. Multiple logistic regression analysis was used to verify the predictors and associations between different study variables. The linear correlation between hepatic and inflammatory markers was evaluated using the Pearson correlation coefficient. Statistical significance was set at a *p* value of <0.05.

## 3. Results

The study participants were mostly women, non-smokers, with more than 6 months of long-term COVID-19, who were not hospitalised, and whose average age was approximately 50 years. Fatigue and dyspnoea were the common symptoms of long-term COVID that affected more than 70% of the patients, followed by muscle weakness, muscle and joint pain, loss of balance, insomnia, chest pain, and cough. The most recurrent comorbidity was arterial hypertension, and other comorbidities were present in 15% or less of the study population (Table 1).

In patients hospitalised during the acute phase of COVID-19, the mean levels of ALT (*p* = 0.0182), AST (*p* = 0.0042), and GGT were high in men (*p* = 0.0024), while that of ferritin were high in both sexes (*p* = 0.0235; *p* = 0.0048). The levels of AST (*p* = 0.0063), LDH (*p* = 0.0024), and ferritin in men were above the reference values (*p* = 0.0350). With regard to the length of time with COVID-19, men in the ≤6 months group had high mean levels of GGT (*p* = 0.0013) and ferritin (*p* = 0.0056). In the ≤6 months group, the levels of GGT (*p* = 0.0192), ferritin (*p* = 0.0084), and total bilirubin (*p* = 0.0370) in men were above the reference values. In the ALT > 29 U/L group, the mean levels of AST (*p* < 0.0001), LDH (*p* = 0.0269), GGT (*p* = 0.0004 and *p* < 0.0001), and ferritin were relatively high in both sexes (*p* = 0.0239 and *p* = 0.0148). The levels of AST (*p* < 0.0001) and GGT were high in both men and women in the ALT > 29 U/L group (*p* = 0.0006 and *p* < 0.0001, respectively,) compared with that of the ALT < 29 U/L group (Table 2).

When the patients were stratified by groups of inflammatory markers (Table 3), we identified high mean levels of ALP (*p* = 0.0092) and GGT in men (*p* = 0.0345), in addition to the higher incidence of ALP > 190 µg/L (*p* = 0.0689). In men with an ESR > 30 mm/h, there was a higher incidence of ALT > 29 U/L (*p* = 0.0370). In women with an ESR > 30 mm/h, the incidence of high mean levels of total bilirubin (*p* = 0.0755) and indirect bilirubin (*p* = 0.0537) were observed. In men with a ferritin level > 300 ng/mL, the mean levels of ALT (*p* = 0.0104), AST (0.0064), and ALP (*p* = 0.0164) increased, in addition to the higher incidence of AST > 25 U/L (*p* = 0.0233). In women with a ferritin level > 300 ng/mL, the mean AST level (*p* = 0.0491) was high.

Age ≥ 60 years (*p* = 0.0214), male sex (*p* = 0.0274), AST > 25 U/L (*p* < 0.0001), and GGT ≥ 50 or 32 U/L (*p* = 0.0019) were associated with ALT > 29 U/L. AST level > 25 U/L was associated with ALT level > 29 U/L (*p* < 0.0001) (Table 4).

Correlation analysis was performed between ALT and AST (*p* < 0.0001, r = 0.670), LDH (*p* = 0.0149, r = 0.1560), GGT (*p* < 0.0001, r = 0.4144), and ferritin (*p* = 0.0006, r = 0.2180) levels in the general study population (Figure 2). In hospitalised patients, a correlation was found between ALT and AST (*p* < 0.0001, r = 0.7502), ALP (*p* = 0.0036, r = 0.3088) and GGT (*p* < 0.0001, r = 0.4772). The other correlations are shown in Figure 3.

## 4. Discussion

In this study, patients with long-term COVID-19 symptoms that lasted up to 20 months were admitted. Alterations were identified in the average levels of ALT, AST, LDH, GGT, and ferritin. The elevation of ferritin, a group of inflammatory markers, was related to the increase in the levels of other liver markers, and that of ESR was above the reference values in all study groups. In addition, age > 60 years, male sex, and AST and GGT levels above the reference values were strongly associated with ALT > 29 U/L. The high correlation between ALT and AST levels also suggests a high risk of liver damage.

An et al. [18] also reported elevated levels of ALT, GGT, and ALP within 14 days after hospital discharge and 2 months following the initial infection. According to Gameil et al. [19], the ALT, AST, GGT, and ALP levels may be elevated 3 months after the resolution of COVID-19. Such altered markers are the result of direct damage to the hepatocytes caused by the virus, and a systemic inflammatory process has already been documented in hospitalised patients with elevated levels of IFN-λ, interleukin IL-6, IL-10, and IL-2 [20].

Patients with severe COVID-19 have elevated levels of AST, ALT, and GGT [21] and reduced albumin levels [22]. In our study, the mean albumin levels remained normal.

AST has also been used as a liver marker. Because it is produced in the muscles, and patients with long-term COVID-19 experience fatigue and muscle weakness as the most frequent symptoms, the underlying hepatic cause of AST elevation remains unclear [23]. However, the correlation and association between ALT and AST identified in our study suggest that true liver injury is the predominant source of AST elevation.

Bende et al. [17] identified liver stiffness and viscosity in patients with post-COVID-19 syndrome, whose rates were significantly higher in those with lung injury during the acute phase and exacerbated clinical manifestations. Liver viscosity appears to be associated with the degree of inflammation and hepatic steatosis [24].

Although ferritin was the only inflammatory marker that differed between the study groups, the mean ESRs were above the reference values in all groups, which could be explained by a probable residual systemic inflammatory response [19].

In hospitalised patients with COVID-19, liver markers demonstrate a correlation between ALT and AST levels; however, this relationship has not been identified between muscle degradation and inflammatory markers. Hence, this finding suggests that liver damage is directly caused by the presence of the virus in the tissue, whose mechanisms have not yet been fully elucidated [25]. In our study, a correlation was found between markers of muscle degradation (LDH) and inflammation (ferritin).

The alterations in liver markers without concomitant elevation of serum total bilirubin levels observed in patients with acute and long COVID-19, and the elevated expression levels of ACE2 in the cholangiocytes, suggest a persistent systemic inflammatory response in these patients [26]. The ACE2-mediated direct viral invasion of the hepatocytes, disrupted immune homeostasis, systemic inflammatory response, concomitant hypotension, pneumonia-associated hypoxia, cytokine storm with increased proinflammatory cytokines, and drug use suggest a pathogenesis of liver injury in patients with long-term COVID-19 [27,28].

An et al. [18] highlighted that patients in a serious condition during acute COVID-19 used more drug therapies, such as oxygen inhalation, antiviral drugs, anti-infective drugs, vasoactive drugs, hormone therapy, immunoregulatory drugs, drugs that regulate the intestinal flora, and symptomatic treatment drugs. Most of these drugs have hepatotoxic effects and may cause liver damage in patients with long-term COVID-19 [29,30].

In addition, reliable control groups are difficult to obtain owing to the ongoing pandemic that affected many people. This study is the first to evaluate the risk of liver damage in patients with long-term COVID-19 of up to 20 months, mainly those living in the Amazon Region.

## 5. Conclusions

Changes in liver function markers, such as ALT, AST, LDH, GGT, and ferritin, may be observed in patients who developed long-term COVID-19, especially those hospitalised during the acute phase of the infection. This may be due to the direct injury caused by the virus to the hepatocytes and a persistent systemic inflammatory process. Furthermore, the results of this study suggest that changes in the markers of liver injury in patients with long-term COVID-19 may persist for more than 1.5 years after the resolution of COVID-19. Finally, new studies should be carried out, especially those that involve the long-term monitoring of these patients, to identify whether such findings are permanent or transient, which represents one of the main gaps in current scientific knowledge.

## Figures and Tables

**Figure 1 ijerph-20-05281-f001:**
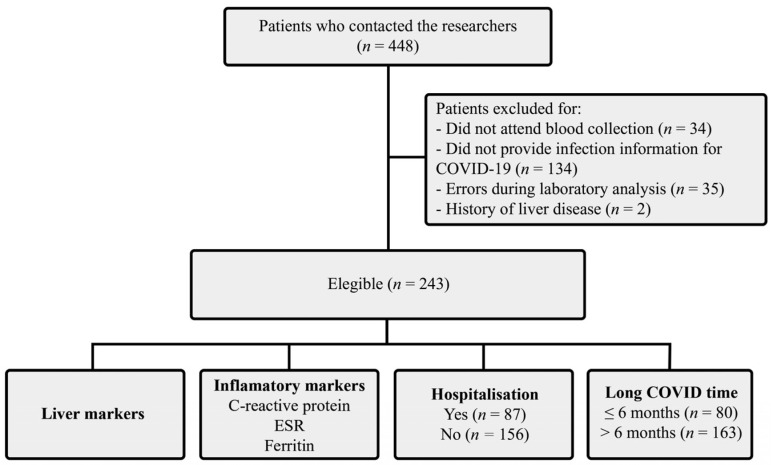
Flowchart of the patient recruitment process. Belém/PA, 2020–2021. ESR: erythrocyte sedimentation rate.

**Figure 2 ijerph-20-05281-f002:**
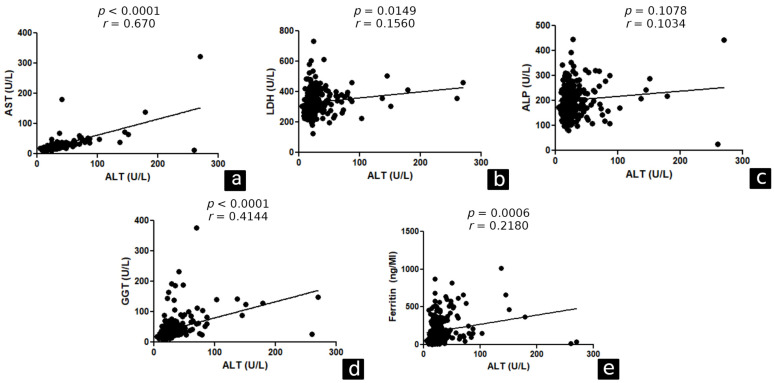
Correlation between ALT and markers of liver function in the general study population. Belém/PA, 2020–2021. ALT: alanine aminotransferase, AST: aspartate aminotransferase, LDH: lactic dehydrogenase, ALP: alkaline phosphatase, GGT: gamma-glutamyl transferase. (**a**) correlation between ALT and AST. (**b**) correlation between ALT and LDH. (**c**) correlation between ALT and ALP. (**d**) correlation between ALT and GGT. (**e**) correlation between ALT and ferritin.

**Figure 3 ijerph-20-05281-f003:**
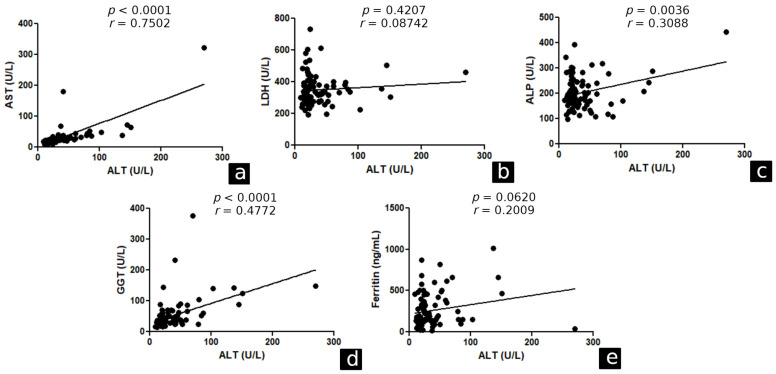
Correlation between ALT and liver function markers in patients who were hospitalised during acute COVID-19. Belém/PA, 2020–2021. ALT: alanine aminotransferase, AST: aspartate aminotransferase, LDH: lactic dehydrogenase, ALP: alkaline phosphatase, GGT: gamma-glutamyl transferase. (**a**) correlation between ALT and AST. (**b**) correlation between ALT and LDH. (**c**) correlation between ALT and ALP. (**d**) correlation between ALT and GGT. (**e**) correlation between ALT and Ferritin.

**Table 1 ijerph-20-05281-t001:** Baseline data of the study patients. Belém/PA, 2020–2021.

Variables	Patients
Women, *n* (%)	159 (65.43)
Age (mean ± SD, years)	49.33 ± 12.83
Smoker/ex-smoker	70 (28.80)
Long COVID symptoms (*n*, %)	
Fatigue	184 (75.72)
Dyspnoea	178 (73.25)
Muscle weakness	164 (67.48)
Muscle and joint pain	152 (62.55)
Loss of balance	146 (52.51)
Insomnia	118 (48.56)
Chest pain	116 (47.73)
Cough	97 (39.91)
Comorbidities (*n*, %)	
Arterial hypertension	80 (32.92)
Respiratory disease	39 (16.04)
DM	19 (7.82)
Heart disease	19 (7.82)
Kidney disease	1 (0.41)
Hospital internment (*n*, %)	87 (35.80)
Up to 30 days	75 (30.86)
>30 days	12 (4.93)
Long COVID time (*n*, %)	
≤6 months	80 (32.92)
>6 months	163 (67.07)

SD: standard deviation, DM: diabetes mellitus, COVID: coronavirus disease.

**Table 2 ijerph-20-05281-t002:** Comparison of liver function laboratory variables between hospitalisation groups, post-COVID time, and ALT > 29 U/L. Belém/PA, 2020–2021.

Variables	Hospitalisation			Long COVID Time			ALT > 29 U/L		
	Yes	No	*p* Value	≤6 Months	>6 Months	*p* Value	Yes	No	*p* Value
ALT, M ± SD	37.68 ± 37.84	27.73 ± 26.42	0.0182 ^&^	31.74 ± 26.86	31.07 ± 33.31	0.5740	-	-	-
ALT > 29 U/L, *n* (%)	33 (13.58)	41 (16.87)	0.0808	25 (10.28)	49 (20.16)	0.9674	-	-	-
AST, M ± SD	30.85 ± 37.55	22.79 ± 12.34	0.0042 ^&^	23.37 ± 9.79	26.80 ± 29.43	0.7790	39.43 ± 41.30	19.65 ± 4.76	<0.0001 ^&^
AST > 25 U/L, *n* (%)	32 (13.16)	31 (12.75)	0.0063 ′	21 (8.64)	42 (17.28)	0.9402	49 (20.16)	14 (5.76)	<0.0001 ′
LDH, M ± SD	348.01 ± 97.94	322.23 ± 66.78	0.1474	345.01 ± 92.30	324.81 ± 72.79	0.0873	343.11 ± 70.56	326.36 ± 83.64	0.0269 ^&^
LDH > 460 U/L, *n* (%)	10 (4.11)	3 (1.23)	0.0024 ′′	7 (2.88)	6 (2.47)	0.1289	3 (1.23)	10 (4.11)	0.7594
ALP, M ± SD	203.10 ± 64.25	201.74 ± 62.18	0.8535	201.81 ± 54.45	202.43 ± 66.67	0.7035	206.32 ± 67.17	200.44 ± 60.91	0.4821
ALP > 190 µg/L, *n* (%)	46 (18.93)	84 (34.56)	0.9908	47 (19.34)	83 (34.15)	0.3110	42 (17.28)	88 (36.21)	0.5932
GGT *, M ± SD	60.15 ± 61.14	37.07 ± 21.46	0.0024 ^&^	64.02 ± 64.78	35.68 ± 15.97	0.0013 ^&^	69.60 ± 68.45	35.04 ± 14.03	0.0004 ^&^
GGT > 50 µg/L, *n* (%)	17 (6.99)	8 (3.29)	0.1371	17 (6.99)	8 (3.29)	0.0192	18 (7.40)	7 (2.88)	0.0006 ′
GGT **, M ± SD	44.47 ± 34.94	39.09 ± 34.82	0.3912	37.02 ± 26.60	44.30 ± 36.46	0.5645	64.97 ± 45.70	32.56 ± 26.14	<0.0001 ^&^
GGT > 32 µg/L, *n* (%)	19 (7.81)	42 (17.28)	0.3773	16(6.58)	45 (18.51)	0.9318	26 (10.70)	35 (14.40)	<0.0001 ′
PT, M ± SD	12.29 ± 1.01	12.87 ± 5.63	0.2142	12.35 ± 1.18	12.82 ± 5.50	0.4390	13.20 ± 8.08	12.43 ± 1.17	0.3034
PT > 15 s, *n* (%)	1 (0.41)	2 (0.82)	1.0000	2 (0.82)	1 (0.41)	0.2528	1 (0.41)	1 (0.41)	0.2937
Ferritin *, M ± SD	338.11 ± 229.71	226.72 ± 155.61	0.0235 ^&^	368.02 ± 238.19	208.27 ± 146.69	0.0056 ^&^	352.28 ± 236.85	239.32 ± 166.66	0.0239
Ferritin > 300 ng/mL, *n* (%)	24 (9.87)	11 (4.52)	0.0350 ′	23 (9.46)	13 (5.35)	0.0084 ′	18 (7.40)	17 (6.99)	0.1904
Ferritin **, M ± SD	177.09 ± 154.13	119.38 ± 116.52	0.0048 ^&^	122.92 ± 110.85	175.07 ± 167.44	0.2090	161.74 ± 133.42	125.81 ± 127.60	0.0148 ^&^
Ferritin > 300 ng/mL, *n* (%)	6 (2.47)	11 (4.52)	0.3907	3 (1.23)	14 (5.76)	0.5628	6 (2.47)	11 (4.52)	0.3691
CRP-positive ***, *n* (%)	10 (4.11)	23 (9.46)	0.6076	9 (3.70)	24 (9.87)	0.5867	11 (4.52)	22 (9.05)	0.8545
ESR *, M ± SD	35.22 ± 23.63	33.10 ± 28.35	0.3698	33.95 ± 23.81	34.31 ± 27.72	0.8629	30.71 ± 21.07	36.75 ± 28.64	0.5343
ESR > 20 mm. *n* (%)	31 (12.75)	22 (9.05)	0.3394	26 (10.69)	27 (11.11)	0.6856	22 (9.05)	31 (12.75)	0.8484
ESR **, M ± SD	41.16 ± 28.33	43.93 ± 25.24	0.4544	51.24 ± 32.88	41.72 ± 26.49	0.1343	48.66 ± 33.94	41.42 ± 22.77	0.4176
ESR > 30 mm, *n* (%)	25 (10.29)	77 (31.68)	0.5882	27 (11.11)	75 (30.86)	0.9403	28 (11.52)	74 (30.45)	0.3403
TB, M ± SD	0.46 ± 0.23	0.47 ± 0.21	0.7087	0.48 ± 0.28	0.46 ± 0.18	0.7587	0.47 ± 0.25	0.47 ± 0.21	0.7099
TB > 1.0 mg/dL	2 (0.82)	1 (0.41)	0.5634	3 (1.23)	0	0.0370 ′′	1 (0.41)	2 (0.82)	0.9756
DB. M ± SD	0.17 ± 0.05	0.17 ± 0.07	0.7317	0.17 ± 0.07	0.17 ± 0.06	0.8638	0.17 ± 0.07	0.17 ± 0.06	0.9341
DB > 0.3 mg/dL, *n* (%)	1 (0.41)	7 (2.88)	0.1369	3 (1.23)	5 (2.05)	0.9980	4 (1.64)	4 (1.64)	0.2326
IB, M ± SD	0.31 ± 0.21	0.31 ± 0.18	0.5453	0.3094 ± 0.23	0.3145 ± 0.17	0.3736	0.30 ± 0.18	0.31 ± 0.19	0.6345
IB > 0.8 mg/dL, *n* (%)	1 (0.41)	1 (0.41)	1.0000	2 (0.82)	0	0.1135	0	2 (0.82)	0.5772
Albumin, M ± SD	4.09 ± 0.34	4.12 ± 0.38	0.2321	3.9969 ± 0.37	4.1677 ± 0.35	0.6949	4.13 ± 0.34	4.10 ± 0.38	0.6542
Albumin > 4.8 g/dL, *n* (%)	0	0	1.0000	0	0	1.0000	0	0	1.0000

* Male. ** Female. *** Qualitative exam. ALT: alanine aminotransferase, AST: aspartate aminotransferase, LDH: lactic dehydrogenase, TB: total bilirubin, DB: direct bilirubin, IB: indirect bilirubin, ALP: alkaline phosphatase, GGT: gamma-glutamyl transferase, CRP, C-reactive protein, PT: prothrombin time. ESR: erythrocyte sedimentation rate. Data are expressed as mean ± standard deviation (M ± SD) and as absolute and relative frequencies. Mann–Whitney test (*p* < 0.05 ^&^), chi-squared test (*p* < 0.05 ′), and Fisher’s exact test (*p* < 0.05 ′′).

**Table 3 ijerph-20-05281-t003:** Comparison of the laboratory variables of liver function status between groups by inflammatory markers. Belém/PA, 2020–2021.

Variables	CRP ***			ESR (mm/h) *			ESR (mm/h) **			Ferritin * (ng/mL)			Ferritin ** (ng/mL)		
	+	−	*p* Value	>20	≤20	*p* Value	>30	≤30	*p* value	>300	≤300	*p* Value	>300	≤300	*p* Value
ALT, M ± SD	43.79 ± 59.36	29.33 ± 23.77	0.1145	35.04 ± 23.19	42.58 ± 41.21	0.7073	27.92 ± 28.78	27.70 ± 34.76	0.4037	49.48 ± 41.58	29.49 ± 16.52	0.0104 ^&^	25.88 ± 10.55	28.07 ± 32.56	0.2520
ALT > 29 U/L *n* (%)	11 (4.52)	63 (25.92)	0.8545	22 (9.05)	13 (5.35)	0.0370 ′	28 (11.52)	11 (4.52)	0.3403	18 (7.40)	17 (6.99)	0.1904	6 (24.69)	33 (13.58)	0.3691
AST, M ± SD	32.33 ± 52.94	24.63 ± 16.53	0.9236	24.37 ± 7.73	34.87 ± 36.62	0.6597	26.00 ± 31.27	21.30 ± 6.77	0.3626	31.83 ± 22.67	25.69 ± 23.82	0.0064 ^&^	24.06 ± 6.25	24.34 ± 26.83	0.0491 ^&^
AST > 25 U/L *n*. (%)	9 (3.70)	54 (22.22)	0.9811	18 (7.40)	10 (4.11)	0.9363	25 (10.29)	10 (41.15)	0.4139	17 (6.99)	11 (4.52)	0.0233 ′	4 (1.64)	31 (12.75)	0.9970
LDH M ± SD	332.06 ± 81.41	331.37 ± 80.10	0.8762	334.15 ± 100.17	331.13 ± 95.05	0.9078	333.13 ± 70.74	326.16 ± 67.18	0.4893	348.03 ± 107.76	322.32 ± 89.52	0.2968	343.76 ± 72.56	329.06 ± 69.06	0.4222
LDH > 460 U/L *n*. (%)	2 (0.82)	11 (4.52)	0.9803	3 (1.23)	4 (1.64)	0.4148	3 (1.23)	3 (1.23)		4 (1.64)	3 (1.23)		1 (0.42)	5 (2.06)	0.4817
ALP M ± SD	232.97 ± 84.13	197.40 ± 57.53	0.0092 ^&^	199.13 ± 63.82	193.77 ± 48.21	0.2808	210.38 ± 68.42	195.12 ± 57.88	0.3712	209.02 ± 53.64	188.67 ± 60.53	0.5312	225.41 ± 59.24	202.46 ± 65.49	0.1520
ALP > 190 µg/L	23 (9.46)	107 (44.03)	0.0689	30 (12.34)	16 (6.58)	0.8287	54 (22.22)	30 (12.34)	0.8980	23 (9.46)	23 (9.46)	0.1383	11 (4.52)	73 (30.04)	0.4349
GGT * M ± SD	49.80 ± 31.46	49.42 ± 49.24	0.9774	52.24 ± 57.38	44.64 ± 26.30	0.8457	-	-	-	66.31 ± 69.34	37.39 ± 16.17	0.0164 ^&^	-	-	-
GGT > 50 µg/L	2 (0.82)	23 (9.46)	0.6297	16 (6.58)	9 (3.70)	0.8923	-	-	-	14 (5.76)	11 (4.52)	0.1356	-	-	-
GGT ** M ± SD	46.25 ± 33.47	39.29 ± 35.11	0.0345 ^&^	-	-	-	40.65 ± 33.79	40.26 ± 36.91	0.3522	-	-	-	56.70 ± 53.14	38.57 ± 31.67	0.1226
GGT > 32 µg/L	15 (6.17)	46 (18.93)	0.1076	-	-	-	39 (16.05)	22 (9.05)	0.8980	-	-	-	9 (3.70)	52 (21.40)	0.2965
PT M ± SD	14.57 ± 11.98	12.36 ± 1.18	0.2444	12.33 ± 1.27	12.66 ± 0.99	0.3124	12.37 ± 1.26	13.50 ± 9.15	0.8224	12.42 ± 1.35	12.47 ± 1.06	0.6394	12.77 ± 1.20	12.78 ± 5.88	0.1465
PT > 15 s	1 (0.42)	2 (0.82)	0.3535	1 (0.42)	0	1.0000	1 (0.42)	1 (0.42)	1.0000	1 (0.42)	3 (1.23)	0.6374	1 (0.42)	2 (0.82)	0.2883
TB M ± SD	0.45 ± 0.21	0.47 ± 0.22	0.8213	0.49 ± 0.26	0.50 ± 0.22	0.9049	0.42 ± 0.19	0.52 ± 0.22	0.0755	0.48 ± 0.27	0.50 ± 0.24	0.7357	0.41 ± 0.19	0.46 ± 0.21	0.4735
TB > 1.0 mg/dL	0	3 (1.23)	1.0000	1 (0.42)	1 (0.42)	1.0000	0	1 (0.42)	0.3390	1 (0.42)	1 (0.42)	1.0000	0	1 (0.42)	1.0000
DB M ± SD	0.16 ± 0.06	0.17 ± 0.06	0.4465	0.17 ± 0.06	0.17 ± 0.06	1.0000	0.16 ± 0.06	0.18 ± 0.07	0.4565	0.16 ± 0.05	0.18 ± 0.06	0.4773	0.14 ± 0.05	0.17 ± 0.07	0.3428
DB > 0.3 mg/dL	1 (0.42)	7 (2.88)	1.0000	2 (0.82)	1 (0.42)	0.5508	3 (1.23)	2 (0.82)	0.9969	0	3 (1.23)	0.5361	0	5 (2.06)	0.5860
IB M ± SD	0.33 ± 0.20	0.31 ± 0.19	0.6692	0.31 ± 0.21	0.32 ± 0.19	0.7328	0.27 ± 0.17	0.37 ± 0.20	0.0537	0.32 ± 0.25	0.31 ± 0.19	0.7899	0.26 ± 0.15	0.33 ± 0.19	0.5066
IB > 0.8 mg/dL	0	2 (0.82)	1.0000	1 (0.42)	1 (0.42)	1.0000	0	0	1.0000	1 (0.42)	1 (0.42)	1.0000	0	0	1.0000
Albumin M ± SD	4.17 ± 0.40	4.09 ± 0.36	0.4643	4.01 ± 0.34	4.13 ± 0.37	0.2978	4.17 ± 0.36	4.07 ± 0.39	0.2630	4.03 ± 0.41	4.06 ± 0.34	0.6314	4.08 ± 0.36	4.15 ± 0.37	0.5909
Albumin > 4.8 g/dL	0	0	1.0000	0	0	1.0000	0	0	1.0000	0	0	1.0000	0	0	1.0000

* Male. ** Female. *** Qualitative exam. ALT: alanine aminotransferase, AST: aspartate aminotransferase, LDH: lactic dehydrogenase, TB: total bilirubin, DB: direct bilirubin, IB: indirect bilirubin, ALP: alkaline phosphatase, GGT: gamma-glutamyl transferase, CRP: C-reactive protein, PT: prothrombin time, ESR: erythrocyte sedimentation rate. Data are expressed as mean ± standard deviation (M ± SD) and as absolute and relative frequencies. Mann–Whitney test (*p* < 0.05 ^&^) and chi-squared test (*p* < 0.05 ′).

**Table 4 ijerph-20-05281-t004:** Factors associated with changes in liver injury markers, post-COVID time, symptoms, and hospitalisation in study patients. Belém/PA, 2020–2021.

Risk Variables	ALT > 29 U/L		AST > 25 U/L		Ferritin > 300 ng/mL		Long COVID Time ≤ 6		>5 Long COVID Symptoms		Hospitalisation	
	Odds Ratio	*p* Value	Odds Ratio	*p* Value	Odds Ratio	*p* Value	Odds Ratio	*p* Value	Odds Ratio	*p* Value	Odds Ratio	*p* Value
Hospitalisation, yes	0.8948	0.8025	2.1662	0.0807	1.7693	0.1495	2.4271	0.0059	1.9802	0.0542	-	-
Long COVID time ≤ 6	0.7877	0.5933	0.6341	0.3236	1.5344	0.2711	-	-	2.8343	0.0042	2.4542	0.0052
Age ≥ 60 years	0.2855	0.0214	1.1367	0.8038	2.4342	0.0331	1.5585	0.2126	0.9407	0.8678	1.4307	0.3218
Male	2.6959	0.0274	0.9369	0.8878	4.8173	<0.0001	2.0594	0.0351	0.6988	0.3046	2.4365	0.0087
>5 long COVID symptoms	1.1528	0.7536	0.6118	0.2554	0.6997	0.3829	2.8768	0.0041	-	-	1.9906	0.0539
ALT > 29 U/L	-	-	21.3046	<0.0001	1.7447	0.2720	0.8798	0.7707	1.0923	0.8321	0.8981	0.8054
AST > 25 U/L	21.5317	<0.0001	-	-	1.2285	0.6807	0.7166	0.4531	0.6006	0.2344	2.2260	0.0670
LDH ≥ 460 U/L	0.2712	0.1577	5.4802	0.0266	1.3118	0.7063	1.2126	0.7673	2.0278	0.3957	4.6938	0.0370
ALP ≥ 190 U/L	0.7481	0.4745	1.0868	0.8380	1.9531	0.0852	1.3307	0.3638	0.7894	0.4398	0.7015	0.2669
GGT ≥ 50 U/L * ou 32 U/L **	3.5989	0.0019	1.9687	0.1094	1.5007	0.3431	1.3967	0.3401	1.7849	0.1055	1.2841	0.4755
Ferritin ≥ 300 ng/mL	1.6449	0.3357	1.3499	0.5338	-	-	1.5394	0.2654	0.6965	0.3639	1.9245	0.0966
ESR ≥ 20 mm/h * ou 30 mm/h **	1.1198	0.7887	1.2075	0.6576	1.9377	0.1069	1.3981	0.3084	0.6022	0.1178	1.0869	0.8008
Positive C-reactive protein ***	1.2672	0.6725	0.8917	0.8434	0.4243	0.1865	0.7255	0.5023	1.1572	0.7462	0.9796	0.9651

* Male. ** Female. *** Qualitative exam. ALT: alanine aminotransferase, AST: aspartate aminotransferase, LDH: lactic dehydrogenase, ALP: alkaline phosphatase, GGT: gamma-glutamyl transferase, ESR: erythrocyte sedimentation rate.

## Data Availability

The data that support the findings of this study are available from the corresponding author, L.F.M.F., upon reasonable request. The data are not publicly available as the information included could compromise the privacy of research participants.

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
