# Peer review of "Liver Function in Patients with Long-Term Coronavirus Disease 2019 of up to 20 Months: A Cross-Sectional Study"

_ijerph, 2023, doi:10.3390/ijerph20075281_

Round 1

Reviewer 1 Report

The definition of “Long COVID-19 is missed in text. Which inclusion criteria did the authors meet to enroll individuals with long COVID-19? This must be clarified in the methods section fully. The only 6-months period of time is not enough.

The time of blood collection is not provided. Were the samples collected in acute phase or post it? Must be defined.

In the results section, the study population is defined as “females”, line 108: “The study participants were composed of female individuals” and then “Women” is a variable in the Table 1 which is confusing and then in the next column the “patients are mentioned as 182 (65,46) and the parenthesis percent must be the females percentage… if yes, the line 108 is not correct!?

In addition to the result section is written really poor and the main points must be mentioned in the text.

Grammatical errors: Line 29 “survive” needs a preposition,  line 56 “all patients provided written”, must be passive voice

Line 42, “CD14+CD16+” both or either?! Should be clarified.

Author Response

Thank you for your contribution, which has improved the quality of this article. And we reinforce that the manuscript was sent to a professional english correction company.

Point 1: The definition of “Long COVID-19 is missed in text. Which inclusion criteria did the authors meet to enroll individuals with long COVID-19? This must be clarified in the methods section fully. The only 6-months period of time is not enough.

Response 1: Thanks for your review. We changed the text as you suggested (line 65-70)

Point 2: The time of blood collection is not provided. Were the samples collected in acute phase or post it? Must be defined.

Response 2: Thanks for your review. We changed the text as you suggested (line 86-89).

Point 3: In the results section, the study population is defined as “females”, line 108: “The study participants were composed of female individuals” and then “Women” is a variable in the Table 1 which is confusing and then in the next column the “patients are mentioned as 182 (65,46) and the parenthesis percent must be the females percentage… if yes, the line 108 is not correct!?

Response 3: Thanks for your review We have made some text adjustment to clarify these results presentation (line 124).

Point 4: In addition to the result section is written really poor and the main points must be mentioned in the text.

Response 4: Thanks for pointing this out. We've made the appropriate changes (127-130).

Point 5: Grammatical errors: Line 29 “survive” needs a preposition,  line 56 “all patients provided written”, must be passive voice.

Response 5: Thanks for your review. We changed the text as you suggested (line 29 and 58-59).

Point 6: Line 42, “CD14+CD16+” both or either?! Should be clarified.

Response 6: Thanks for pointing this out. We've made the appropriate changes (line 43).

Reviewer 2 Report

Given that liver damage is potentially associated with COVID-19, it is important to investigate the effect of COVID-19 on liver function. Usually, it will be more meaningful to do a longitudinal trajectory analysis; for example, examining liver function every 6 months. In this study, Lima and coworkers examined the hepatic clinical-laboratory profile of patients with up to 20 months of long-term COVID-19. Some questions and concerns are listed below.

1. The manuscript is not well laid out, particularly the result section. For example, the description of Table 3 (lines 131-136) should be separated from the legend of Table 2 (lines 127-130).

2.  243 patients were eligible for this study and they were allocated into 4 groups. Please indicate how many people participated in each group in Figure 1. In addition, please also show how many people are infected for more than 6 months.

3.  Did the authors exclude the participants who have chronic hepatitis B/C virus infection?

4.  In this study, the variables have been set to specific values. For example, 29 for ALT, 25 for AST, 460 for LDH, and so on. Actually, the normal range of ALT is around 7-35 U/L in women and 7-40 U/L in men and different countries have different setting ranges. Please explain why you set these values when comparison of liver function.     

5. Some drugs may affect liver function. Did the participants take any hepatoprotective drugs or anti-inflammatory drugs during the study?

Author Response

Thank you for your contribution, which has improved the quality of this article. And we reinforce that the manuscript was sent to a professional English correction company.

Point 1: The manuscript is not well laid out, particularly the result section. For example, the description of Table 3 (lines 131-136) should be separated from the legend of Table 2 (lines 127-130).

Response 1: We are grateful for your. We have made some text adjustment (line 147-150 and 158).

Point 2: 243 patients were eligible for this study and they were allocated into 4 groups. Please indicate how many people participated in each group in Figure 1. In addition, please also show how many people are infected for more than 6 months.

Response 2: Thanks for your review. We have made some text and image adjustments to clarify these in methods presentation (line 73-79).

Point 3: Did the authors exclude the participants who have chronic hepatitis B/C virus infection?

Response 3: Thanks for your review. We have made some text adjustment to clarify these in methods presentation (line 72).

Point 4: In this study, the variables have been set to specific values. For example, 29 for ALT, 25 for AST, 460 for LDH, and so on. Actually, the normal range of ALT is around 7-35 U/L in women and 7-40 U/L in men and different countries have different setting ranges. Please explain why you set these values when comparison of liver function.

Response 4: Thanks for your review. We have made some text adjustment to clarify these methods presentation (line 109-111). additionally we added supplementary material with the reference values adopted by the laboratory (Supplementary Table S1)

Point 5: Some drugs may affect liver function. Did the participants take any hepatoprotective drugs or anti-inflammatory drugs during the study?

Response 5: Thanks for your review. We have made some text adjustment to clarify these methods presentation (line 64).

Round 2

Reviewer 1 Report

The changes are satisfying.